# Cloud model-based evaluation of landslide dam development feasibility

**Dengze Luo[1,2], Hongtao Li[1,2], Yu Wu[1,2], Dong Li[1,2], Xingguo Yang[1,2], Qiang Yao[1,2]***

**1** State Key Laboratory of Hydraulics and Mountain River Engineering, Sichuan University, Chengdu, Sichuan, China, **2** College of Water Resource and Hydropower, Sichuan University, Chengdu, Sichuan, China

\* yaoqiang777@126.com

## Abstract

As natural backwater structures, landslide dams both threaten downstream human settlement or infrastructure and contain abundant hydro-energy and tourism resources, so research on their development feasibility is of great significance for permanently remedying them and effectively turning disasters into benefits. Through an analysis of the factors influencing landslide dam development and utilization, an index system (consisting of target, rule, and index layers) for evaluating development feasibility was constructed in this paper. Considering uncertainty and randomness in development feasibility evaluation, a cloud model-improved evaluation method was proposed to determine membership and score clouds based on the uncertainty reasoning of cloud model, and a cloud model-improved analytic hierarchy process (AHP-Cloud Model) was introduced to obtain weights. Final evaluation results were obtained using a hierarchical weighted summary. The improved method was applied to evaluate the Hongshiyan and Tangjiashan landslide dams and the results were compared with the maximum membership principle results. The results showed that the cloud model depicted the fuzziness and uncertainty in the evaluation process. The improved method proposed in this paper overcame the loss of fuzziness in the maximum membership principle evaluation results, and was capable of more directly presenting evaluation results. The development feasibility of the Hongshiyan landslide dam was relatively high, while that of the Tangjiashan landslide dam was relatively low. As suggested by these results, the evaluation model proposed in this paper has great significance for preparing a long-term management scheme for landslide dams.

## 1 Introduction

Landslide dams are natural dams formed when the barrier bodies produced by earthquakes, landslides, debris flows, volcanic eruption, or other geological action which are dammed in gullies, river channels, or depressions. In recent years, Southwest China has experienced active geological tectonic movements, frequent earthquakes, and heavy rainfall [1], resulting in the emergence of hundreds of landslide dams. For instance, the Tangjiashan landslide dam formed by the 2008 Wenchuan earthquake has a maximum storage capacity of $3.02\times10^8$ m$^3$ [2]; the Hongshiyan landslide dam created by the 2014 Ludian earthquake has a maximum storage capacity of $2.6\times10^8$ m$^3$ [3]; and the Baige landslide dam caused by landslides on the banks of

**Data Availability Statement:** All relevant data are within the manuscript.

**Funding:** This research was funded by the National Key R&D Program of China, grant number 2018YFC1508501 (https://service.most.gov.cn/)

and National Natural Science Foundation of China, grant number No. 51809188 (http://www.nsfc.gov.cn/). The funders had no role in study design, data collection and analysis, decision to publish, or preparation of the manuscript.

**Competing interests:** The authors have declared that no competing interests exist.

the Jinsha River in Tibet in 2018 has a maximum storage capacity of $2.9 \times 10^8$ m$^3$ [4]. Such natural landslide dams usually have no drainage gallery, and the banked-up water level seriously threatens the safety of life and property in both upstream and downstream areas, turning landslide dams into a major concern for natural disaster prevention and control [5]. On the other hand, as a consequence of natural feedbacks to riverbed incision, landslide dams not only promote healthy river development [6], but also present irrigation, power generation, tourism, and other potential [7–9]. With respect to the development of landslide dams, there are already some precedents both at home and abroad. For example, the Xiaonanhai landslide dam in Chongqing, formed by the 1856 earthquake, has now been transformed into a multi-purpose water conservancy project mainly used for irrigation and urban water supply, as well as power generation, tourism, and aquaculture [10]. The landslide that occurred in Quarto, Sul Savio in 1812 created a landslide dam on the Savio River, which was ultimately transformed into an impounding dam for hydropower in 1923 [11]. The Waikarernoan landslide dam in New Zealand has also been successfully used for hydropower generation, with an installed capacity of $12.4 \times 10^4$ kW [12]. However, precedents like these all involved landslide dams that had been safe for hundreds of years before transformation. Thus, for the recently-formed landslide dams that were retained after emergency management, it is urgently necessary to evaluate their development feasibility, and identify scientific long-term management schemes.

Over the past few years, global scholars have introduced many methods to evaluate landslide dams from multiple aspects. Ermini et al. [13] statistically analyzed 84 natural landslide dams, developed an experience-based geomorphologic dimensionless blockage index (DBI) method, and selected dam volume, catchment area, and dam height as evaluation indices to predict dam stability. Xu et al. used a fuzzy mathematical method to evaluate the risk grade of the Hongshiyan landslide dam, and established six main indices (i.e., social development; dam material, volume parameters, water level growth rate, mountain stability and river channel river) for rating risk [1]. Based on 43 landslide dam cases from Japan, Dong et al. defined significant variables influencing stability through discriminant analysis, constructed AHWL and PHWL multiple regression models with strong predictive ability [14], and optimized the two models using Logistic regression and jack-knife techniques [15]. Zhang et al. evaluated the stability of the Hongshiyan landslide dam in the emergency management stage using rapid stability evaluation methods, and discussed their development feasibility for management from the perspective of economic benefits [8]. Frigerio et al. derived a Bayesian model based on the input of missing observations of a landslide dam, aiming to predict its service life [16]. Relying on data from all over the world., Shen et al. created a landslide dam database and built a regression model for predicting service life based on regression analysis [17]. However, existing studies have typically focused on predicting dam stability and service life in the initial emergency management stage, but have rarely considered development feasibility after emergency management. Studies have pointed out that the development of a landslide dam must ensure its safety and health, and consider environmental compatibility, social benefits, and economic benefits, but they have failed to come up with any specific evaluation model or method. Current evaluation methods are largely statistical analysis-based or fuzzy methods. While methods based on statistical models must be supported by a large number of samples, however, the scarcity of engineering practice in the development of landslide dams makes it difficult to produce a large sample set. On the other hand, the fuzzy evaluation method lacks a model for the conversion between qualitative and quantitative relationships, not to mention that the concept of fuzziness is no longer fuzzy when a membership function is used to describe the fuzzy sets [18]. Therefore, a cloud model which can transform between qualitative concept and quantitative value by combining the fuzziness and randomness is more appropriate to evaluate the development feasibility of landslide dams [19].

Given the lack of studies devoted to landslide dam development feasibility and the defects of existing evaluation methods, this paper examined multiple factors influencing their development and constructed an evaluation index system using top-down hierarchical decomposition. Relying on cloud model uncertainty reasoning, it proposed a cloud model-improved evaluation method to describe the uncertainty and randomness in the evaluation process. The method was then applied to the Hongshiyan and Tangjiashan landslide dams, and evaluation results were verified according to the maximum membership principle. The results suggest that the evaluation method and index system proposed in this paper have important guiding significance for evaluating landslide dam development and preparing a long-term management scheme.

## 2 Construction of evaluation index system

### 2.1 Evaluation indices

The indices constituting a feasibility evaluation system must be able to reflect their overall characteristics and influencing factors, while being independent of each other and easy to obtain. Decisions about landslide dam development and utilization are influenced by multiple factors. Taking Hongshiyan landslide dam as an example, emergency measures are taken to reduce the breaching probability after its formation, and the potential hydropower energy, economic benefits and other conditions are taken into account to evaluate the potential for development based on the safety of the landslide dam [8]. So, referring to the hierarchical logic of the AHP target, rule, and index layers, this paper examined landslide dam risk ratings and the development feasibility evaluation of hydropower projects from existing studies, investigated the characteristics and influencing factors of landslide dam development and utilization, and divided landslide dam development feasibility targets after emergency management into four rule layers, i.e., safety risk, resource feasibility, economic feasibility, and eco-environmental impact.

Safety risk: landslide dam development and utilization must, first of all, be safe [20]. Currently evaluations are carried out mainly in two aspects, i.e., stability and the safety risk of their failure. According to existing studies [14, 21–23], the factors influencing landslide dam stability include: (1) the geomorphologic geometry, (2) the inflow rate, and (3) the material composition and geological structure. Thus, DBI(dimensionless blockage index), a volume parameter for barrier bodies, is used to evaluate stability under geomorphologic geometry [13]. The structure and material composition of barrier bodies are used to reflect the influence of dam material properties and grain composition on their stability [11]. Given that a dam is subjected to overtopping and failure when the inflow rate exceeds the discharge rate [3], the standard of flood control for discharge structures is used to measure barrier body discharge capacity. Considering that a landslide dam failure causes massive casualties and property losses [1], two measurement indices have been introduced: population at risk and important downstream towns and public infrastructure.

Resource feasibility: the water head elevated by a landslide dam can be used as hydro-energy for power generation, and stored water resources can be used for irrigation. The storage capacity of a natural reservoir formed by a landslide dam both satisfies the demand for power system regulation and reflects its flood control capacity. In addition, due to geological tectonic movement, landslide dam formation is accompanied by the emergence of lakes, which have tourism development value. For this reason, hydro-energy resources, tourism, irrigation area, and natural reservoir regulation capacity are indices used to measure the landslide dam resource feasibility.

Economic feasibility: according to standard international procedures, the primary consideration in landslide dam development is its power generation capacity [8]. Thus, economic

feasibility is examined from the perspective of power generation benefits, and cost per kilo-watt-hour and installed capacity are selected for quantification.

Eco-environmental impact: landslide dam development has both positive and negative impacts on the eco-environment. On one hand, hydropower, as a clean renewable resource, has the positive benefits of energy conservation and emissions reduction; on the other hand, permanent landslide dam development negatively impacts local species and further exacerbates soil erosion. Therefore, energy conservation benefit, emissions reduction benefit, soil erosion impacts, and species impacts are used to evaluate eco-environmental impact.

The index system for evaluating landslide dam development feasibility was created as shown in Table 1.

## 2.2 Standard for evaluation

First, landslide dam development feasibility was measured at four levels, i.e., high, relatively, relatively low, and relatively low. The next step was to create the set $V = \{v1, v2, v3, v4\}$ = {grade I, grade II, grade III, and grade IV} = {high feasibility, relatively feasibility, relatively low feasibility, and low feasibility}, and the ranges of various evaluation indices were determined. Specifically, for the volume parameter, "DBI>3.08" was defined as low feasibility, whereas "DBI<2.75" was defined as high feasibility; the range between them was divided into two intervals, which corresponded to relatively low feasibility and relatively high feasibility [1, 13]. Following the rule of "higher risk, lower feasibility", the ranges of indices such as material composition, population at risk, and important infrastructure were determined according to relevant domestic standards [24]. The indices for coal conservation rate and carbon dioxide

**Table 1. Index system for evaluating landslide dam development feasibility.**

| | Indicator layer | Indicator | Description |
|---|---|---|---|
| Index system for evaluating landslide dam development feasibility | Safety risk A1 | DBI (dimensionless blockage index) B1 | Reflects the stability of the landslide dam in its natural state |
| | | The structure and material composition of barrier bodies B2 | Reflects the permeability of the dam |
| | | Population at risk B3 | Reflects the threat to the lives of people downstream of dam failure. |
| | | Important downstream towns and public infrastructure B4 | Reflects the condition of critical infrastructure in the area affected by the dam failure |
| | | The standard of flood control for discharge structures B5 | Reflects the drainage capability of discharge structures |
| | Resource feasibility A2 | Tourism resource B6 | Reflects the tourism resources of the landslide dam and its surroundings |
| | | Irrigation area B7 | Reflects the potential irrigation area of the surrounding area |
| | | Hydro-energy resources B8 | Reflects the hydro-energy potential of the basin |
| | | Natural reservoir regulation capacity B9 | Reflects the regulating capacity of natural reservoirs formed by weir lakes. |
| | Economic feasibility A3 | Unit energy investment B10 | Reflects the cost per kilowatt-hour in planning stage |
| | | Installed capacity B11 | Reflects the total installed capacity of the dam in planning stage |
| | Eco-environmental impact A4 | Energy conservation benefit B12 | Reflects the benefits of replacing coal power with hydropower |
| | | Emissions reduction benefit B13 | Reflects the ability to reduce $CO_2$ emissions |
| | | Soil erosion impacts B14 | Reflects the impact of erosion |
| | | Species impacts B15 | Reflects impacts on local flora and fauna diversity |

**Table 2. Standards for evaluation of landslide dam development feasibility.**

| Indicator | High feasibility | Relatively feasibility | Relatively low feasibility | Low feasibility |
|---|---|---|---|---|
| | (IV) | (III) | (II) | (I) |
| DBI(dimensionless blockage index) | >3.08 | 3.08–2.92 | 2.92–2.75 | <2.75 |
| The structure and material composition of barrier bodies | Soil dominated by earth | Earth with boulders | Boulders with earth | Boulders dominated by earth |
| Population at risk($10^4$person) | >100 | 100–10 | 10–1 | <1 |
| Important downstream towns and public infrastructure | Nationally important facilities or large water projects Facilities of provincial importance | Facilities of provincial importance | Facilities of municipal importance | Facilities of general importance and the following |
| The standard of flood control for discharge structures (year) | 2–5 | 5–10 | 10–20 | 20–50 |
| Tourism resource | Deficient | Relatively deficient | Relatively rich | Rich |
| Irrigation area(mu) | <0.1 | 0.1–1 | 1–30 | >30 |
| Hydro-energy resources(MW) | 10–100 | 100–200 | 200–400 | >400 |
| Natural reservoir regulation capacity | Daily regulation reservoir | Monthly regulation reservoir | Seasonal regulation reservoir | Annual or multi-year regulation of reservoirs |
| Unit energy investment(yuan/kW) | 5.5.-5 | 5–4.5 | 4.5–4 | <4 |
| Installed capacity (MW) | 0. 5 | 0.5–25 | 25–250 | >250 |
| Energy conservation benefit(t/kW) | 0.4–0.5 | 0.5–0.6 | 0.6–0.7 | >0.7 |
| Emissions reduction benefit(kg/$m^3$) | <1 | 1–2 | 2–3 | 3–4 |
| Soil erosion impacts | Serious | Strong | Medium | Slight |
| Species impacts | Serious | Strong | Medium | Slight |

reduction rate were measured with reference to the *Standard for evaluation of green small hydropower stations* [25]. The measurement of other indices referred to literature in related fields. Standards for defining indices such as flood control, hydro-energy resources, and installed capacity were established, and qualitative indices were verbally described after consulting experts in related fields. See specific ranges in Table 2.

## 3 Evaluation methods

### 3.1 Cloud model

The cloud model, proposed by Li et al. [26] and based on probability statistics and fuzzy mathematics, is a model for converting between qualitative concepts and quantitative values. It describes the randomness and fuzziness of fuzzy concepts and overcomes the deficiencies of traditional evaluation methods in dealing with randomness and fuzziness, which has been extensively used in the fields of multi-criteria group decision making, risk evaluation, and so forth [19, 27]. Assuming that *U* is a universe of discourse expressed by an exact numerical magnitude and that *C* is a qualitative concept within *U*, then, for element *x* within any universe of discourse, there is a random number $u(x)\in[0,1]$ with stabilization bias, which is referred to as the membership of *x* relative to *C*. In this case, the distribution of *x* within *U* is called a cloud, and each *x* value is referred to as a cloud droplet, or a quantitative description of qualitative concepts. The numerical magnitude of *u(x)* reflects the representation of qualitative concept *C* by the corresponding cloud droplet, so the closer the value of *u(x)* is to 1, the stronger the ability of the corresponding cloud droplet *x* will embody the overall characteristics of the qualitative concept [26]. The cloud model describes the numerical characteristics of a fuzzy concept using expected value *Ex*, entropy *En*, and hyper-entropy *He*, where expected value *Ex* denotes the mean value of the qualitative concept, corresponding to the central position of a cloud droplet; entropy *En* denotes the fuzziness of the qualitative concept (or the discreteness

of the corresponding cloud droplet relative to the mean value), and reflects the acceptable numerical range; hyper-entropy *He* is the entropy of the *En*, reflects the uncertainty of entropy (or the cohesion of the corresponding cloud droplet), and corresponds to the thickness of the cloud layer [18]. In this paper, interval number [*CL*, *CR*] was used to define indices under various evaluation grades, where a boundary value is a transition point between two adjacent grades and belongs to the two grades at the same time. That is, the boundary value corresponding to two adjacent grades has a membership of 0.5 [27]. We deduced the formula for the characteristic parameters of the cloud model (see Formula (1)) to convert qualitative concepts into three numerical characteristics.

$$
\begin{cases}
Ex = (C_L + C_R)/2 \\
En = (C_L - C_R)/2.355 \\
He = c \bullet En
\end{cases}
\tag{1}
$$

Where hyper-entropy *He* ranges from 0~*En*, and reflects the uncertainty of the index [26]. It was set as $c = 0.1$ in this paper. For an index whose interval number is a unilateral set, if [-∞, $C_L$] or [$C_R$, +∞], *Ex* was as a boundary value $C_L$ or $C_R$, *En* was set as *En*-1, and a half normal cloud model was used for expression [19].

In the cloud model, the conversion between qualitative concepts and quantitative values requires aid from forward, backward, and conditional cloud generators [26]. The forward cloud generator outputs three numerical characteristics (expected value *Ex*, entropy *En*, and hyper-entropy *He*) for the input qualitative concept as N cloud droplets $(x, u(x))$, completing the conversion of range and distribution rules from qualitative concepts to quantitative data in linguistic value expressions (Fig 1(A)). The backward cloud generator is based on mathematical statistics, and expresses a qualitative concept by extracting the three numerical characteristics of the cloud model from a certain number of cloud droplets (Fig 1(B)). Conditional cloud generators include *X* and *Y*. Given the three numerical characteristics and specific value $x_0$ of a cloud, the former generates cloud droplet $u(x_0)$ corresponding to the membership of $x_0$; given the three numerical characteristics and membership $u(y_0)$ of a cloud, the latter generates a cloud droplet corresponding to the specific value $y_0$ of $u(y_0)$. They can be combined for the uncertainty reasoning of the cloud model, thus converting between qualitative concepts and quantitative values. The specific flow is shown in Fig 1(C). Table 3 shows the basic operations of cloud.

### 3.2 Uncertainty reasoning of cloud model

The uncertainty reasoning of cloud model is based on knowledge about uncertainty and adopts the format of "if A then B" to express qualitative concept relationships. A,

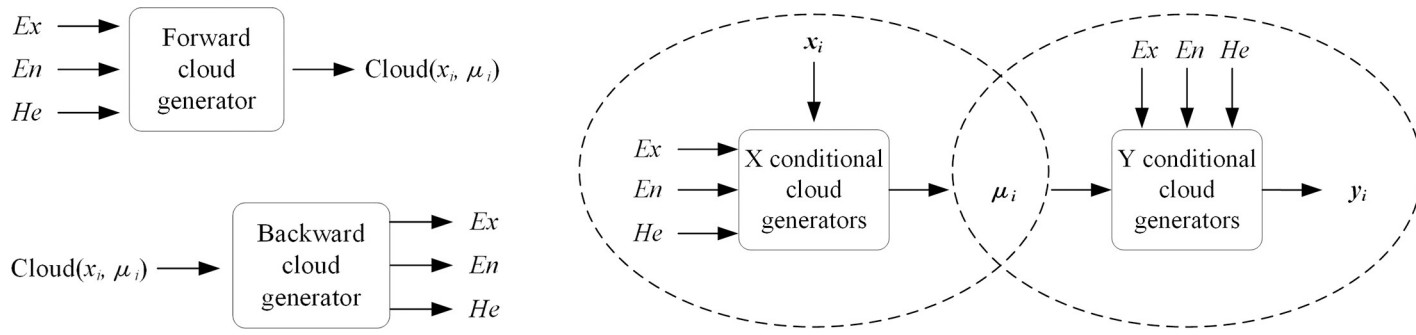

**Fig 1. Three kinds of cloud generator.** (a) Forward cloud generator. (b) Backward cloud generator. (c) Conditional cloud generator.

**Table 3. Basic operations of cloud.**

| arithmetic symbol | Cloud model digital characteristics algorithm | | |
|---|---|---|---|
| | $Ex$ | $En$ | $He$ |
| + | $Ex_1+Ex_2$ | $\sqrt{En_1{}^2+En_2{}^2}$ | $\sqrt{He_1{}^2+He_2{}^2}$ |
| - | $Ex_1-Ex_2$ | $\sqrt{En_1{}^2+En_2{}^2}$ | $\sqrt{He_1{}^2+He_2{}^2}$ |
| × | $Ex_1{\times}Ex_2$ | $\lvert Ex_1Ex_2\rvert \cdot \sqrt{\left(\frac{En_1}{Ex_1}\right)^2+\left(\frac{En_2}{Ex_2}\right)^2}$ | $\lvert Ex_1Ex_2\rvert \cdot \sqrt{\left(\frac{He_1}{Ex_1}\right)^2+\left(\frac{He_2}{Ex_2}\right)^2}$ |
| ÷ | $Ex_1{\div}Ex_2$ | $\lvert\frac{Ex_1}{Ex_2}\rvert \cdot \sqrt{\left(\frac{En_1}{Ex_1}\right)^2+\left(\frac{En_2}{Ex_2}\right)^2}$ | $\lvert\frac{Ex_1}{Ex_2}\rvert \cdot \sqrt{\left(\frac{He_1}{Ex_1}\right)^2+\left(\frac{He_2}{Ex_2}\right)^2}$ |

corresponding to X conditional cloud generator, is referred to as the antecedent of the rule, while B, corresponding to Y conditional cloud generator, is referred to as the consequent. In combination they constitute cloud model-based rule generators [28], as shown in Fig 1.

**3.2.1 Building an antecedent cloud model.** Table 2 provides the evaluation indices for landslide dam development feasibility and grade. For quantitative indices, an antecedent cloud model was constructed according to Formula (1) and the ranges of various indices in Table 2. For qualitative indices, first they were converted into interval numbers [0, 1], [1, 2], [2, 3], and [3, 4] according to grades corresponding to qualitative comments, and then these interval numbers were further converted into cloud model numbers according to Formula (1): $C_4$(0.5, 0.4246, 0.0425), $C_3$(1.5, 0.4246, 0.0425), $C_2$(2.5, 0.4246, 0.0425), and $C_1$(3.5, 0.4246, 0.0425).

**3.2.2 Building a consequent cloud model.** The hundred-mark system was adopted to score landslide dam development feasibility so that the higher the feasibility, the higher the score. Scores were measured at four levels, i.e., "low, relatively low, relatively high, and high". They corresponded to interval numbers of [0, 25], [25, 50], [50, 75], and [75, 100], which were converted into cloud model numbers: $C_4$(12.5, 10.62, 1.06), $C_3$(37.5, 10.62, 1.06), and $C_2$(62.5, 10.62, 1.06), $C_1$(87.5, 10.62, 1.06). See the conversion relationships in Table 4:

In the process of uncertainty reasoning, for a given antecedent qualitative concept comment or quantitative value $xi$, the antecedent cloud model can be used to generate the membership $u$ of each grade. Controlled by membership $u$, the cloud model generated a score cloud with uncertainty that represents the consequent qualitative concept, thus propagating uncertainty.

## 3.3 Calculating cloud model-improved AHP

In the process of constructing a judgment matrix according to the traditional analytic hierarchy process (AHP), the scales for importance comparison were definite values, which neither accurately reflected the subject preferences of decision makers nor objectively presented the fuzziness and randomness of the comparison. When calculating weights, adopting algebraic operations to approximately aggregate the opinions of multiple experts can neglect the fuzziness, randomness, and discreteness of expert opinions [29]. In the cloud model-improved

**Table 4. Uncertainty reasoning cloud conversion relationship.**

| Antecedent cloud model | | | Consequent cloud model | | |
|---|---|---|---|---|---|
| Qualitative indicator | Interquartile range | Digital character | Fuzzy concept | Score intervals | Digital character |
| (I) | [0, 1] | (0.5, 0.4246, 0.0425) | High marks | [75, 100] | (87.5, 10.62, 1.06) |
| (II) | [1, 2] | (1.5, 0.4246, 0.0425) | Higher score | [50, 75] | (62.5, 10.62, 1.06) |
| (III) | [2, 3] | (2.5, 0.4246, 0.0425) | Lower score | [25, 50] | (37.5, 10.62, 1.06) |
| (IV) | [3, 4] | (3.5, 0.4246, 0.0425) | Low score | [0, 25] | (12.5, 10.62, 1.06) |

AHP, the cloud model established scales for pairwise importance comparison and endowed the randomness of comparison. Multiple expert opinions were synthesized by floating cloud preference aggregation, thus obtaining objective results with randomness. The scales for pairwise importance comparison were established with reference to the division method proposed by Jiang and Yan et al. [18, 30]. N experts were invited from related fields to score according to the table of scales for importance comparison, thus obtaining n judgment matrices. The opinions of multiple experts were aggregated according to Formula (2), (3) and (4) through floating cloud preference aggregation and produce the final judgment matrix.

$$Ex = \alpha_1 Ex_1 + \alpha_2 Ex_2 + \cdots \alpha_n Ex_n \tag{2}$$

$$En = \frac{\alpha_1 Ex_1 En_1 + \alpha_2 Ex_2 En_2 + \cdots \alpha_n Ex_n En_n}{\alpha_1 Ex_1 + \alpha_2 Ex_2 + \cdots \alpha_n Ex_n} \tag{3}$$

$$He = \sqrt{He_1^2 + He_2^2 + \cdots He_n^2} \tag{4}$$

where the magnitudes of $\alpha_1, \alpha_2, \ldots \alpha_n$ reflect the weights of expert opinions, and satisfy $\alpha_1 + \alpha_2 + \ldots + \alpha_n = 1$.

Adopting the final judgment matrix, the square root method (Formula (5), (6) and (7)) was employed to calculate the cloud model for index weights:

$$Ex_i^{(0)} = \frac{Ex_i}{\sum Ex_i} = \frac{\left(\prod_{j=1}^{n} Ex_{ij}\right)^{\frac{1}{n}}}{\sum_{i=1}^{n}\left(\prod_{j=1}^{n} Ex_{ij}\right)^{\frac{1}{n}}} \tag{5}$$

$$En_i^{(0)} = \frac{En_i}{\sum En_i} = \frac{\left(\left(\prod_{j=1}^{n} Ex_{ij}\right)\sqrt{\sum_{j=1}^{n}\left(\frac{En_{ij}}{Ex_{ij}}\right)^2}\right)^{\frac{1}{n}}}{\sum_{i=1}^{n}\left(\left(\prod_{j=1}^{n} Ex_{ij}\right)\sqrt{\sum_{j=1}^{n}\left(\frac{En_{ij}}{Ex_{ij}}\right)^2}\right)^{\frac{1}{n}}} \tag{6}$$

$$He_i^{(0)} = \frac{He_i}{\sum He_i} = \frac{\left(\left(\prod_{j=1}^{n} Ex_{ij}\right)\sqrt{\sum_{j=1}^{n}\left(\frac{He_{ij}}{Ex_{ij}}\right)^2}\right)^{\frac{1}{n}}}{\sum_{i=1}^{n}\left(\left(\prod_{j=1}^{n} Ex_{ij}\right)\sqrt{\sum_{j=1}^{n}\left(\frac{He_{ij}}{Ex_{ij}}\right)^2}\right)^{\frac{1}{n}}} \tag{7}$$

The expected values of the calculated cloud model for index weights were checked for consistency, after which normalization processing was performed to obtain the final relative weights of various indices.

## 3.4 Comment set cloud model

Using the cloud model to describe the comment set of landslide dam development feasibility not only achieved boundary fuzzification, but also fully considered model randomness and

**Table 5. Comment set cloud model.**

| Feasibility level | Score intervals | Digital character | | |
|---|---|---|---|---|
| | | *Ex* | *En* | *He* |
| Low feasibility | 0–25 | 0 | 21.2314 | 2.1231 |
| Relatively low feasibility | 25–50 | 37.5 | 10.6157 | 1.0616 |
| Relatively feasibility | 50–75 | 62.5 | 10.6157 | 1.0616 |
| High feasibility | 75–100 | 100 | 21.2314 | 2.1231 |

discreteness, thus reducing the subjective uncertainty in the comparison of evaluation results and better adapting to the language habits of mankind [29]. First, the hundred-mark system was used to depict the scores of various grades, and then they were converted into the characteristic parameters of the cloud model according to Formula (1). For the left and right boundaries, *En* was tripled to guarantee a membership of 0.5 at the comment division boundary, as shown in Table 5. See comment clouds in Fig 2.

### 3.5 Cloud model-improved evaluation steps

The cloud model evaluated landslide dam development feasibility and determined the memberships of qualitative and quantitative indices. Cloud model-improved AHP was employed to determine weights and synthesize the cloud model results. The results were substituted into the comment set cloud model to obtain final landslide dam development feasibility. See the specific flow in Fig 3.

The process was expressed as follows:(1) Convert interval numbers into the numerical characteristics of the cloud model using formula (1) based on the ranges of indices; (2) Determine the score cloud relative to each concept according to qualitative index comments or quantitative index values based on the uncertainty reasoning of cloud model;(3) Calculate the weight cloud distribution through cloud model-improved AHP based on expert evaluation results; (4) Synthesize the operation formula and determine the membership S through evaluation based on cloud operation rules (Table 3);

$$S_i = W_i \bullet R_i \tag{8}$$

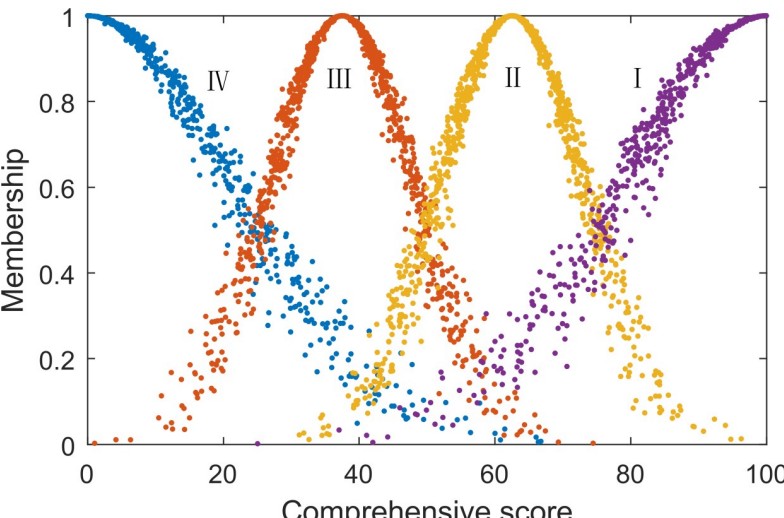

**Fig 2. Comment set cloud model.**

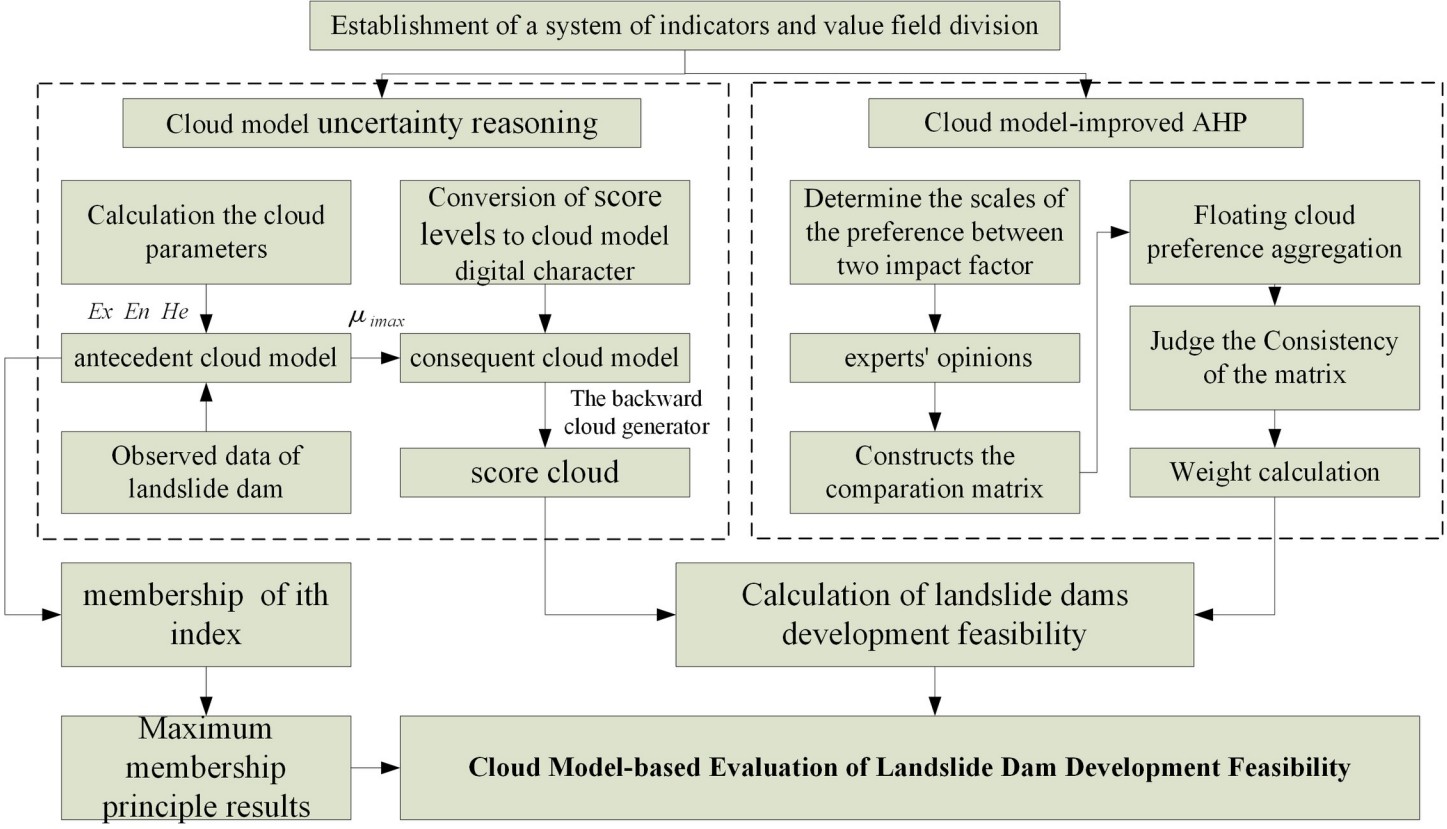

**Fig 3. Flow of cloud model-improved evaluation.**

(5) Substitute weight cloud and various index score values based on cloud model operation rules, and obtain the evaluation result cloud through hierarchical weighted summary.

## 4 Case study

### 4.1 Evaluation of Hongshiyan landslide dam development feasibility

**4.1.1 Project overview.** The Hongshiyan landslide dam in Yunnan Province was formed by landslides on both banks of the Niulan River as a result of the 2014 Ludian earthquake, as shown in Fig 4. The landslide dam has an accumulation height of 103 m and a total volume of about 10 million m³, and controls a drainage area of 12,087 km². Its average annual discharge, Normal storage level, and storage capacity are 127 m³/s, 1,200 m, and 141 million m³, respectively. The resulting reservoir has seasonal regulation functions. The barrier bodies are mainly composed of gravel soil intermingled with isolated and crushed stones, and contain large isolated stones with a maximum diameter of 15 m. The deposits are dense, without hollowing. A dam failure would directly impact 1,015 people distributed in two towns of upstream Huize County, and more than 30,000 people and 30,000 mu farmlands distributed in downstream Ludian, Qiaojia, and Zhaoyang Counties. It would also endanger the upstream Xiaoyantou hydropower station and the downstream Tianhuaban and Huangjiaoshu hydropower stations, etc. After the flood season is over without incident, the landslide dam will be transformed through long-term management into a hydropower station, with a designed installed capacity of 201 MW and an annual power output of 789 million kW·h. Table 6 provides the values of various indices for the Hongshiyan landslide dam, assigned with reference to engineering data.

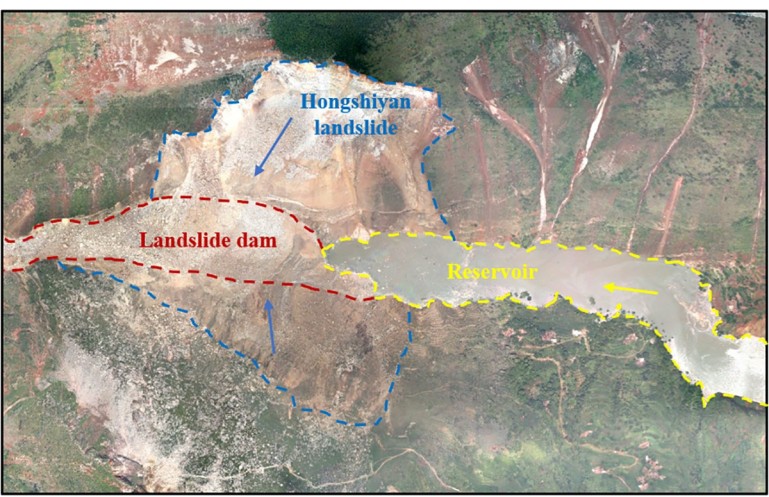

**Fig 4. Aerial view of the landslide dam.**

**4.1.2 Membership and score values.** Relying on cloud model uncertainty reasoning, this paper used the cloud generator to calculate index memberships of indices used the consequent cloud generator to obtain corresponding score clouds, as shown in Table 7:

**4.1.3 Determination of weights.** Weights were calculated through cloud model-improved AHP, as detailed in Table 8. From the aspect of the indicator layer, the safety risk has the greatest weight, which reflects the opinion of safety is the basis of the landslide dam development [20]. While the weight of eco-environmental impact is bigger than the economic feasibility and resource feasibility, indicating that the environmental protection plays a more critical role in the development of landslide dam. The close weighting of economic feasibility and resource feasibility indicates that they are of similar importance.

**4.1.4 Analysis of evaluation results.** After determining the index weight distribution and their memberships relative to comment grade, evaluation results were calculated according to hierarchical weighted summary, as shown in Table 9.

The cloud model-improved method results were substituted into the comment set cloud model, as shown in Fig 5. An expected value of 71.0561 suggested that the development feasibility of the Hongshiyan landslide dam was relatively high. The cloud atlas illustrated that the result cloud fell between "relatively high feasibility" and "high feasibility", but were closer to the former. The entropy and hyper-entropy values of the cloud were 2.0801 and 2.0008, respectively, which were low overall. Reflected in the cloud atlas, they were concentrated in

**Table 6. Values of various indices of the Hongshiyan landslide dam.**

| Indicator | Data | Indicator | Data |
|---|---|---|---|
| DBI(dimensionless blockage index) | 4.9222 | Tourism resource | Relatively rich |
| The structure and material composition of barrier bodies | Boulders with earth | Unit energy investment | 3.99 |
| Population at risk | 3 | Installed capacity | 201 |
| Important downstream towns and public infrastructure | Facilities of municipal importance | Energy conservation benefit | 1.21 |
| The standard of flood control for discharge structures | 20 | Emissions reduction benefit | 3.45 |
| Hydro-energy resources | 127 | Soil erosion impacts | slight |
| Irrigation area | 3.6 | Species impacts | strong |
| Natural reservoir regulation capacity | Seasonal regulation reservoir | | |

**Table 7. Membership and score values.**

| Indicator | Membership | | | | Score cloud | | |
|---|---|---|---|---|---|---|---|
| | (IV) | (III) | (II) | (I) | *Ex* | *En* | *He* |
| DBI(dimensionless blockage index) | 1 | 7.13E-97 | 3.4E-111 | 5.33E-35 | 0 | 0 | 0 |
| The structure and material composition of barrier bodies | 2.95E-05 | 0.0433 | 0.9802 | 0.1001 | 65.0434 | 1.7799 | 1.8102 |
| Population at risk | 0.4419 | 0.3949 | 0.8042 | 0.9654 | 84.1212 | 1.6928 | 1.6989 |
| Important downstream towns and public infrastructure | 0.1050 | 0.9769 | 0.0409 | 2.94E-05 | 34.7338 | 4.2994 | 4.7697 |
| The standard of flood control for discharge structures | 1.55E-26 | 8.6E-07 | 0.4957 | 0.4924 | 77.6751 | 2.6219 | 2.4108 |
| Tourism resource | 1.04E-08 | 0.0001 | 0.0991 | 0.9828 | 85.1226 | 2.4084 | 2.4271 |
| Irrigation area | 9.91E-05 | 5.14E-11 | 0.6212 | 0.5562 | 50.0139 | 3.6222 | 3.6256 |
| Hydro-energy resources | 3.52E-05 | 0.1050 | 0.7831 | 0.6425 | 5.4E+01 | 1.5635 | 1.5562 |
| Natural reservoir regulation capacity | 1.5E-05 | 0.0286 | 0.9375 | 0.1389 | 67.0781 | 1.7079 | 1.7109 |
| Unit energy investment | 0.0120 | 0.0804 | 0.4704 | 0.9997 | 87.4010 | 1.3237 | 1.2743 |
| Installed capacity | 1.67E-54 | 2.31E-43 | 0.7979 | 0.9666 | 84.1744 | 1.3632 | 1.5607 |
| Energy conservation benefit | 9.34E-44 | 2.2E-34 | 3.28E-25 | 1 | 100 | 1.9578 | 2.1639 |
| Emissions reduction benefit | 8.43E-10 | 1.49E-05 | 0.0259 | 0.7189 | 97.8949 | 2.7652 | 2.7180 |
| Soil erosion impacts | 1.34E-08 | 0.0001 | 0.1072 | 0.9750 | 84.6311 | 1.8053 | 1.7471 |
| Species impacts | 0.0646 | 0.9997 | 0.0702 | 6.8E-05 | 37.7888 | 1.7270 | 1.7300 |

distribution, suggesting that the overall evaluation results had relatively low randomness and uncertainty.

## 4.2 Evaluating Tangjiashan landslide dam development feasibility

**4.2.1 Project overview.** The Tangjiashan landslide dam is both the largest and most dangerous landslide dam caused by the 2008 Wenchuan earthquake. In the earthquake, landslides blocked the channel of the Tongkou River, forming the landslide dam about 6 km away in Beichuan County upstream of the Jianhe River. The landslide dam is 82.65~124.4 in height, and 20.37 million m$^3$ in volume. It has a storage capacity of 145 million m$^3$, an upstream catchment

**Table 8. Calculation results of weights.**

| Indicator layer | | | | Indicator | | | |
|---|---|---|---|---|---|---|---|
| Indicator number | *Ex* | *En* | *He* | Indicator number | *Ex* | *En* | *He* |
| A1 | 0.4488 | 0.5026 | 0.4767 | B1 | 0.1310 | 0.1244 | 0.1266 |
| | | | | B2 | 0.3089 | 0.3023 | 0.3128 |
| | | | | B3 | 0.3459 | 0.3429 | 0.3454 |
| | | | | B4 | 0.0537 | 0.0533 | 0.0655 |
| | | | | B5 | 0.1605 | 0.1770 | 0.1497 |
| A2 | 0.1496 | 0.1448 | 0.1401 | B6 | 0.1707 | 0.1691 | 0.1717 |
| | | | | B7 | 0.0784 | 0.0806 | 0.0808 |
| | | | | B8 | 0.4182 | 0.4275 | 0.4236 |
| | | | | B9 | 0.3326 | 0.3227 | 0.3238 |
| A3 | 0.1457 | 0.1414 | 0.138 | B10 | 0.6796 | 0.8093 | 0.75 |
| | | | | B11 | 0.3204 | 0.1907 | 0.25 |
| A4 | 0.2852 | 0.3093 | 0.2967 | B12 | 0.1733 | 0.1151 | 0.1406 |
| | | | | B13 | 0.1317 | 0.1326 | 0.1281 |
| | | | | B14 | 0.3295 | 0.3538 | 0.3314 |
| | | | | B15 | 0.3656 | 0.3985 | 0.3999 |

**Table 9. Landslide dam development feasibility results.**

| Indicator layer | Evaluation results | | |
|---|---|---|---|
| | *Ex* | *En* | *He* |
| A1 | 63.5247 | 5.6029 | 5.8928 |
| A2 | 63.1794 | 4.9277 | 4.9381 |
| A3 | 86.3673 | 1.9001 | 2.0148 |
| A4 | 71.9174 | 4.2096 | 4.2562 |
| Results | 71.0561 | 2.0801 | 2.0008 |

area of 3,550 km$^2$, and an average annual discharge of 104 m$^3$/s. The barrier bodies are mainly composed of the isolated stone, crushed stone, and residual eluvial gravel soil formed through the squeezing or collapse of bedrock, and the silty fine sand deposited in the reservoir area of the Kuzhu dam. From top down there are four layers, i.e., gravel soil layer, crushed stone layer, isolated crushed stone layer (stratiform-like), and dark gray silty gravel layer. In the downstream areas of the Tangjiashan landslide dam there are many important cities, such as Mianyang, Santai, and Suining. If it were to burst, the lives and safety of more than 1.3 million people living in these areas would be gravely threatened. According to plans for the Tongkou River basin, if the Tangjiashan landslide dam were transformed into a hydropower station, it would have an installed capacity of 110 MW and an annual power output of 500 million kWh. The detailed information about Tangjiashan landslide dam is shown in Table 10.

**4.2.2 Membership and score values.** The memberships of various indices relative to different grades and their quantitative values were obtained based on cloud model uncertainty reasoning, as shown in Table 11:

**4.2.3 Evaluation results.** The development feasibility results for the Tangjiashan landslide dam were obtained according to hierarchical weighted summary, as shown in Table 12:

The cloud model-improved method results were substituted into the comment set cloud model, as shown in Fig 6. An expected value of 46.54 suggested that the development feasibility of the Tangjiashan landslide dam was relatively low. The cloud atlas illustrated that the result

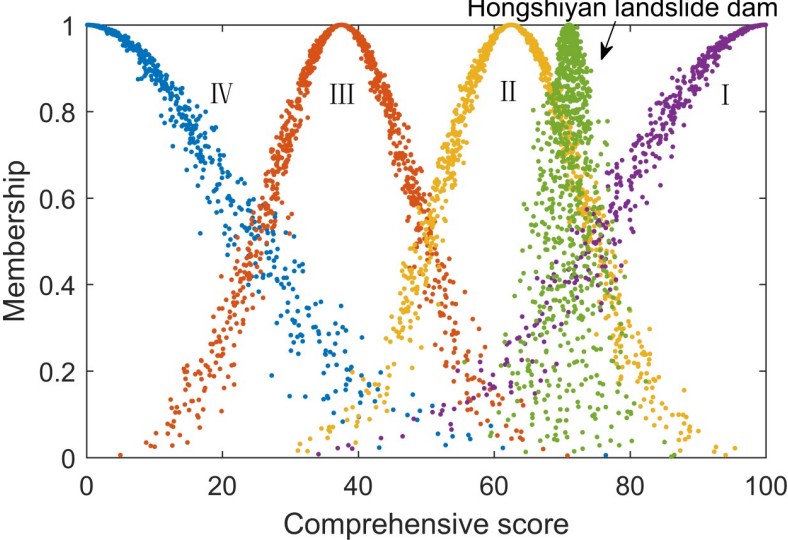

**Fig 5. Hongshiyan landslide dam development feasibility results.**

**Table 10. Indices values of the Tangjiashan landslide dam.**

| Indicator | Data | Indicator | Data |
|---|---|---|---|
| DBI(dimensionless blockage index) | 4.15 | Tourism resource | rich |
| The structure and material composition of barrier bodies | Boulders with earth | Unit energy investment | 5.14 |
| Population at risk | 120 | Installed capacity | 110 |
| Important downstream towns and public infrastructure | Nationally important facilities or large water projects Facilities of provincial importance | Energy conservation benefit | 1.4 |
| The standard of flood control for discharge structures | 20 | Emissions reduction benefit | 5.43 |
| Hydro-energy resources | 196.6 | Soil erosion impacts | medium |
| Irrigation area | 10 | Species impacts | strong |
| Natural reservoir regulation capacity | Annual regulation reservoirs | | |

cloud was closer to the comment grade of relatively low feasibility. The entropy and hyper-entropy values were 2.0862 and 2.0079, respectively, which were low overall. Reflected in the cloud atlas, they were concentrated in distribution, suggesting that the overall evaluation results had relatively low randomness and uncertainty.

## 5 Discussion

### 5.1 Calculation and verification of the maximum membership principle

The level corresponding to the element with the largest membership degree in the fuzzy evaluation vector is taken as the evaluation result, which is the maximum membership principle [29]. In the calculation process based on cloud model uncertainty reasoning, the antecedent cloud generator obtained the memberships of various indices relative to different grades by inputting qualitative comments or quantitative values, as shown in Tables 7 and 11. Studies have indicated that when the cloud model method is used as a substitute for traditional membership functions in determining memberships, it can fully consider the uncertainty mapping between evaluation indices and sets, depict the fuzziness and randomness between them, and

**Table 11. Membership and score values.**

| Indicator | Membership | | | | Score cloud | | |
|---|---|---|---|---|---|---|---|
| | (IV) | (III) | (II) | (I) | *Ex* | *En* | *He* |
| DBI(dimensionless blockage index) | 1 | 5.65E-46 | 2E-56 | 1.92E-16 | 0 | 0 | 0 |
| The structure and material composition of barrier bodies | 0.0002 | 0.1570 | 0.9163 | 0.0247 | 57.1793 | 0.5356 | 0.0021 |
| Population at risk | 0.9652 | 0.1477 | 4.7E-139 | 7.74E-33 | 0 | 0 | 0 |
| Important downstream towns and public infrastructure | 0.9948 | 0.0838 | 0.0001 | 7.19E-09 | 13.7916 | 0.129 | 0.011 |
| The standard of flood control for discharge structures | 8.24E-27 | 4.87E-07 | 0.4963 | 0.4934 | 77.5513 | 1.4891 | 0.2608 |
| Tourism resource | 4.67E-09 | 6.47E-05 | 0.0683 | 0.9999 | 87.4426 | 0.0055 | 0.0001 |
| Irrigation area | 2.36E-25 | 2.62E-78 | 0.9023 | 0.7155 | 56.7369 | 0.5616 | 0.0507 |
| Hydroenergy resources | 0.0019 | 0.5397 | 0.4697 | 0.4780 | 51.6901 | 1.4165 | 0.18 |
| Natural reservoir regulation capacity | 2.48E-09 | 4.84E-05 | 0.0543 | 0.9948 | 88.8342 | 0.1304 | 0.0098 |
| Unit energy investment | 0.8704 | 0.1866 | 0.0004 | 0.0308 | 19.2370 | 0.6753 | 0.0867 |
| Installed capacity | 4.79E-18 | 2.83E-14 | 0.9585 | 0.7598 | 58.7717 | 0.3824 | 0.0435 |
| Energy conservation benefit | 3.78E-70 | 3.16E-53 | 9.44E-42 | 1 | 100 | 0.1 | 0.01 |
| Emissions reduction benefit | 3.47E-14 | 6.99E-09 | 0.0001 | 1 | 100 | 0.1 | 0.01 |
| Soil erosion impacts | 6.39E-05 | 0.0656 | 0.9999 | 0.0667 | 62.6342 | 0.0127 | 0.0012 |
| Species impacts | 0.2701 | 0.7522 | 0.0105 | 3.67E-06 | 27.8670 | 0.9606 | 0.1509 |

**Table 12. Tangjiashan landslide dam development feasibility results.**

| Indicator layer | Evaluation results | | |
|---|---|---|---|
| | *Ex* | *En* | *He* |
| A1 | 30.8511 | 5.3410 | 5.6425 |
| A2 | 70.5450 | 4.9277 | 4.9381 |
| A3 | 31.9030 | 1.9004 | 2.0148 |
| A4 | 61.3201 | 4.2097 | 4.2562 |
| Results | 46.5363 | 2.0862 | 2.0079 |

achieve more reliable membership results [31]. In this paper, membership calculated by the antecedent cloud model was used to calculate the evaluation results for landslide dam development feasibility according to Formula 2 under the maximum membership principle, as shown in Table 13:

According to the maximum membership principle, the evaluation results for the Hongshiyan landslide dam indicated that its membership relative to the first-level comment was the maximum, so development feasibility was relatively high. Results for the Tangjiashan landslide dam showed that its membership values relative to the second-level and fourth-level comments were both near to 0.35, corresponding to relatively low feasibility and relatively high feasibility. This suggests that development of Tangjiashan landslide dam is feasible, but limited by some factors. The maximum membership principle results were consistent with the cloud model results applied in this paper, suggesting that this method was effective.

When the fuzzy comprehensive evaluation method was used, the results for Tangjiashan landslide dam showed similar memberships for comments of two grades; when the maximum membership was used to determine the final results, it caused a fuzziness loss that ultimately jeopardized the accuracy of the final judgment results [29]. Relative to the maximum membership principle, the cloud model method presented the quantitative evaluation results as three numerical cloud characteristics representing the central value, fuzziness, and randomness. The results had richer contents and were directly presented in the cloud atlas of the comment set for visualization and remedying the defects of the maximum membership principle.

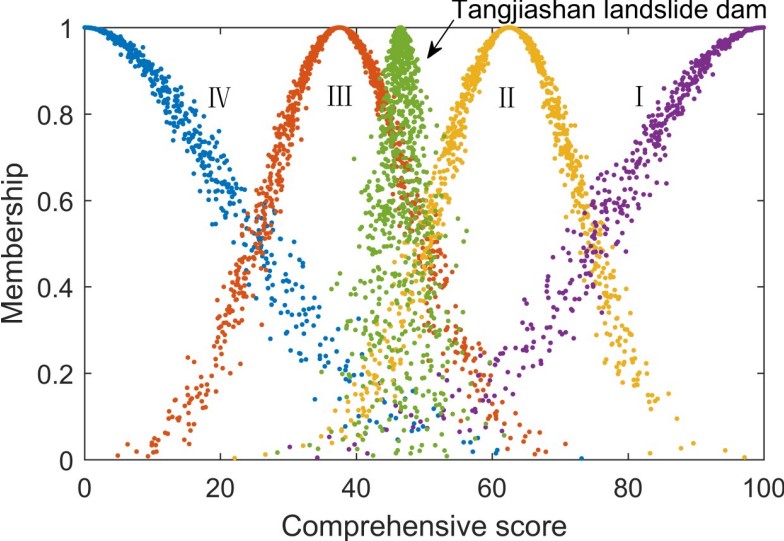

**Fig 6. Tangjiashan landslide dam development feasibility results.**

**Table 13. Maximum membership principle results.**

| Indicator layer | The membership of Hongshiyan landslide dam | | | | The membership of Tangjiashan landslide dam | | | |
|---|---|---|---|---|---|---|---|---|
| | (IV) | (III) | (II) | (I) | (IV) | (III) | (II) | (I) |
| A1 | 0.2895 | 0.2024 | 0.6628 | 0.4439 | 0.5183 | 0.1041 | 0.3627 | 0.0868 |
| A2 | 0.0000 | 0.0535 | 0.7050 | 0.5263 | 0.0008 | 0.2257 | 0.2969 | 0.7576 |
| A3 | 0.0081 | 0.0547 | 0.5754 | 0.9891 | 0.5915 | 0.1268 | 0.3073 | 0.2644 |
| A4 | 0.0236 | 0.3655 | 0.0644 | 0.5892 | 0.0988 | 0.2966 | 0.3333 | 0.3270 |
| Results | 0.1378 | 0.2111 | 0.5052 | 0.5902 | 0.3471 | 0.1835 | 0.3470 | 0.2841 |

## 5.2 Discussion of evaluation results

For the Hongshiyan landslide dam, an analysis and the cloud atlas of the secondary evaluation results (Fig 7) showed that its safety risk A1, resource feasibility A2, economic feasibility A3, and eco-environmental impact A4 corresponded to respective comments of relatively high feasibility, relatively high feasibility, high feasibility, and relatively high feasibility. These comments were consistent with the corresponding comments under the maximum membership principle, suggesting that the Hongshiyan landslide dam is relatively suitable for development in all relevant aspects. At present, the Hongshiyan landslide dam has been transformed into a large-scale pivotal water conservancy project through long-term management, undertaking multiple engineering tasks such as power generation, water supply, and irrigation [8].

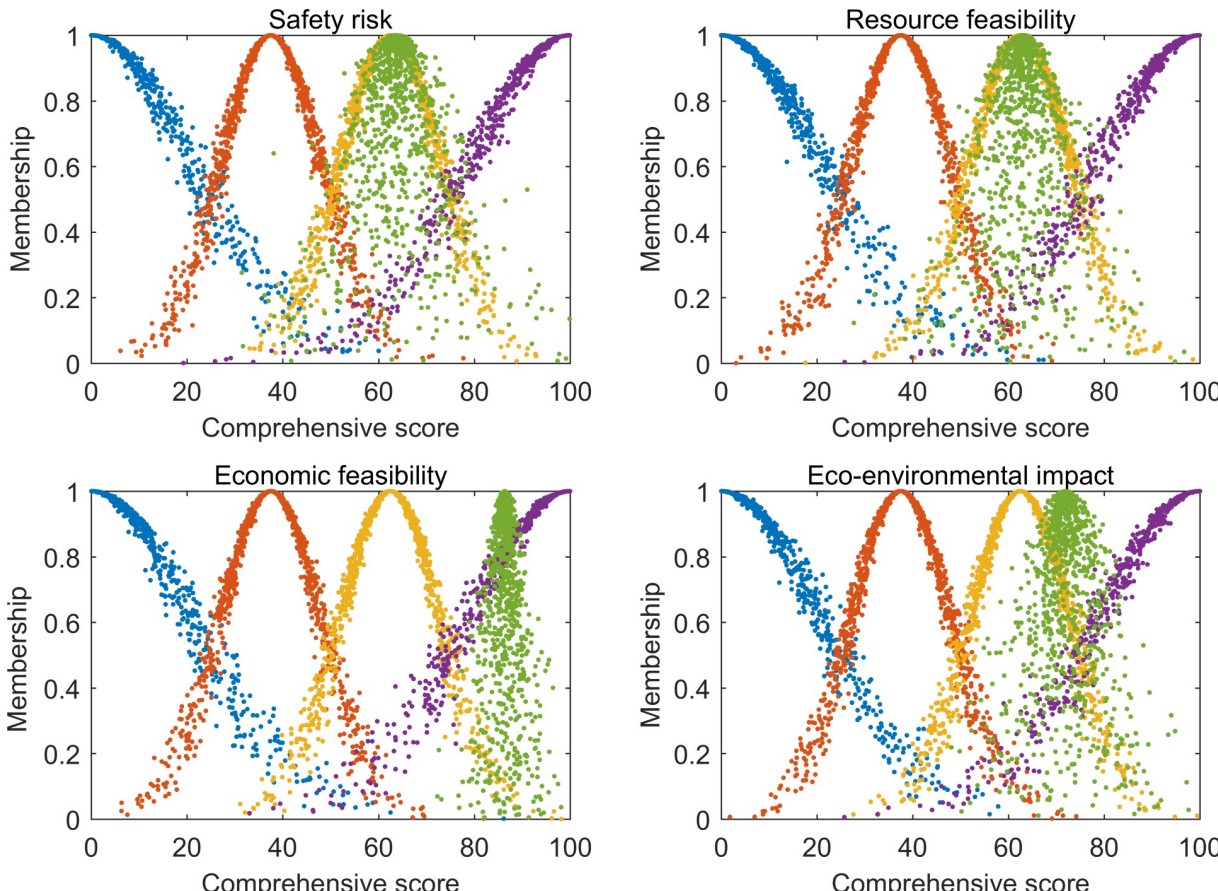

**Fig 7. Cloud atlas for secondary evaluation results of Hongshiyan landslide dam development feasibility.**

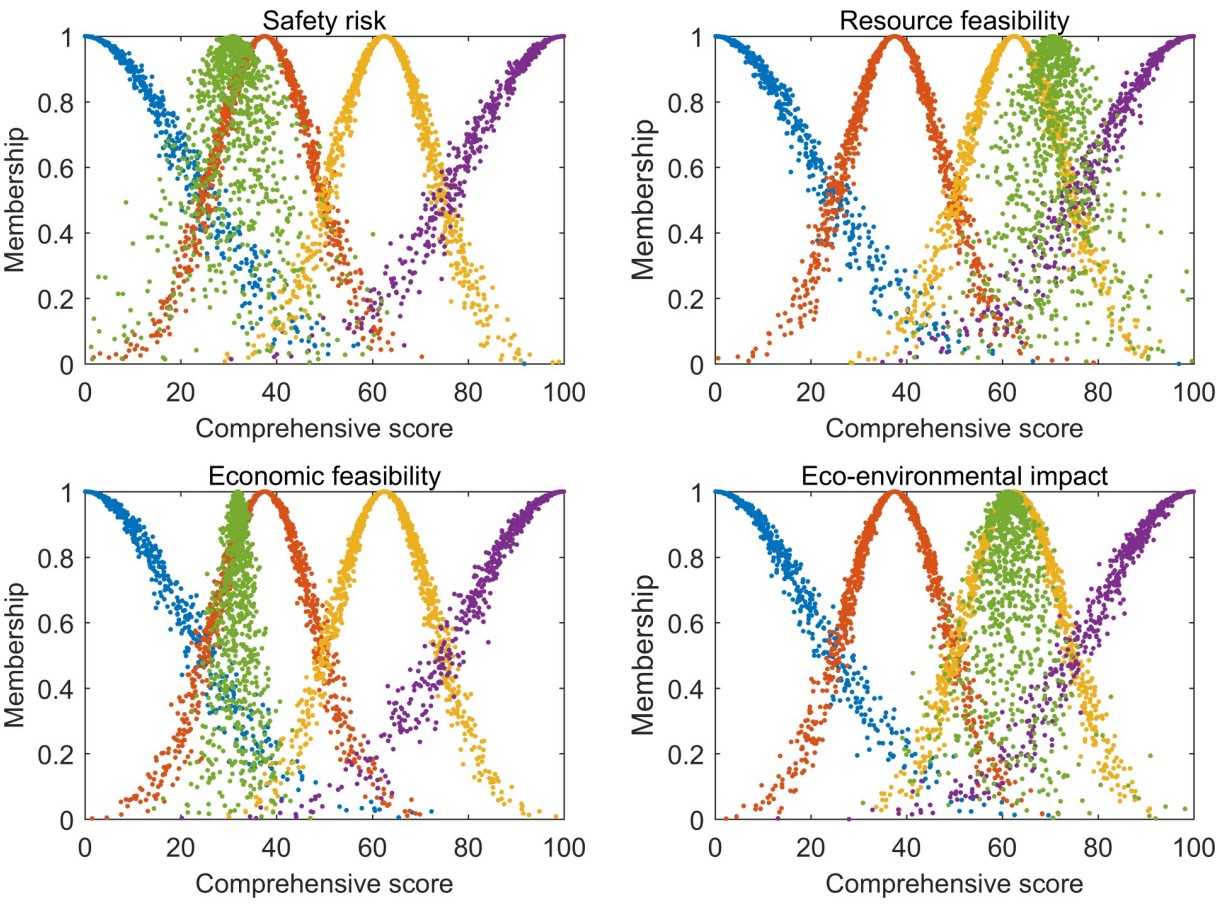

**Fig 8. Cloud atlas for secondary evaluation results of Tangjiashan landslide dam development feasibility.**

As shown in Fig 8, the secondary evaluation indices A1~A4 for the Tangjiashan landslide dam corresponded to respective comments of relatively low feasibility, relatively high feasibility, relatively low feasibility, and relatively high feasibility. The factors limiting its development feasibility were engineering safety risk and economic feasibility. According to further index layer analysis, in terms of economic feasibility, the Tangjiashan landslide dam has a relatively high cost per kilowatt-hour, which negatively influences economic feasibility. However, it also threatens the lives and safety of people living downstream, so long-term management is required. If the strategy is to completely remove the landslide dam, the tremendous dam volume will incur an extremely high engineering cost. On the contrary, if the landslide dam is retained for further development, a built-up landslide dam could not only create tourism, irrigation, electricity generation, and other economic benefits [8], but also increase the social benefits, promoting the local socio-economic development. Therefore, despite the relatively low economic feasibility of the Tangjiashan landslide dam, its potential national social benefits are still considerable. Considering the Tangjiashan landslide dam hasn't been developed, it is hard to measure its social benefits currently, thus in this paper, we only choose economic benefits as evaluation indicator, a more comprehensive evaluation of the landslide dam development feasibility can be conducted combining with social benefits in the future. In terms of engineering safety risk, the main factors obstructing the development of Tangjiashan landslide dam are the landslide dam volume parameter, population at risk, and downstream important infrastructure. Downstream of the Tangjiashan landslide dam there are many small and medium-sized

cities like Mianyang and Jiangyou, so that once it bursts, it would devastate the infrastructure of these cities, and threaten the lives and safety of more than 1.3 million people [32]. Thus, if the strategy is to retain the Tangjiashan landslide dam, engineering measures should be taken to ensure its safety. The countermeasures should focus on reducing the risk of landsliding, piping and overtopping of the dam. Vibrating compaction can be used to improve dam stability, combining with the slope revetment in the upstream and rock pile pressure on the slope toe of the downstream on the basis of knowledge of the physical properties of dam materials. Grouting engineering measures can deal with potential leakage issues. For example, an engineering sealant measure was applied to the Xiaonanhai land-slide dam with excellent sealant effectiveness according to leakage observation in later periods [8]. Besides, widening and lining the existed spillway combining with setting crown wall on the dam top are useful to prevent overtopping. Factors affecting the development of the landslide dam can also be reduced by increasing the downstream flood protection capacity through the construction of facilities such as flood barriers. Currently, after emergency management, the total water storage of the Tangjiashan landslide dam declined from $2×10^8$ m$^3$ to $86×10^6$ m$^3$, which significantly lowered the probability of a Tangjiashan landslide dam burst [8]. As for follow-up development, further planning is under way. The results of this paper were consistent with the results of the two engineering measures that were adopted, suggesting the results were accurate and reliable.

Notably, DBI was selected in this paper to calculate the volume parameters for two landslide dams. According to the study by Ermini et al. [13], the DBIs of both were greater than 3.08, so they were both instable natural dams. Other quantitative and qualitative rapid evaluation methods have also shown that the Hongshiyan landslide dam should be deemed instable [8]. However, the Hongshiyan landslide dam exists between an artificial dam and a hydropower station, and there is a discharge tunnel connecting the natural reservoir and the downstream hydropower station, which means the discharge tunnel can quickly drain water inflow from upstream and prevent overtopping failure. This would provide time for spillway construction and other emergency management measures [3]. For the Tangjiashan landslide dam, an artificial spillway was built within seven days and six nights at the beginning of its formation. The spillway soon controlled the water level and cleared the threat to downstream areas, laying a foundation for long-term management [32]. The above two cases suggest that although natural landslide dams may be judged as instable in rapid evaluations, engineering measures can be taken to strengthen the stability of barrier bodies, reduce the potential risk of dam burst, and turn disasters into benefits to mankind.

## 6 Conclusions

This paper constructed a hierarchical index system for evaluating landslide dam development feasibility from four aspects, i.e., safety risk, resource feasibility, economic feasibility, and eco-environmental impact. Based on a cloud model, an improved evaluation method was proposed and applied to the case studies of the Hongshiyan and the Tangjiashan landslide dams. It was validated using the maximum membership principle. According to the findings of this paper:

1. The method based on cloud model uncertainty reasoning can satisfactorily depict the fuzziness and uncertainty in the evaluation process. It presents the evaluation results as the three numerical cloud characteristics representing the central value, fuzziness, and randomness. The evaluation results had richer contents and were directly presented in the cloud atlas of the comment set for visualization.

2. Following the maximum membership principle, the evaluation method proposed in this paper was validated and the consistency between the two calculation methods suggested

that the method proposed was effective. Compared to the maximum membership principle, the cloud model overcame the fuzziness loss of evaluation results under the highest grade.

3. The Hongshiyan landslide dam evaluation showed relatively high development feasibility, which now has been developed and put into use. While the Tangjiashan landslide dam had low development feasibility, the factors restricting its development are engineering safety risks and economic feasibility, engineering measures like grouting, vibrating compaction and lining the existed spillway can be applied to reducing the safety risks, so as to turn disasters into benefits to mankind. The consistency between the evaluation results and engineering practice verified the reliability of the method proposed in this paper, suggesting that it has great significance for preparing a long-term management scheme for landslide dams.

## Author Contributions

**Conceptualization:** Xingguo Yang.

**Data curation:** Dengze Luo, Yu Wu, Dong Li.

**Funding acquisition:** Qiang Yao.

**Investigation:** Dengze Luo.

**Methodology:** Dengze Luo, Hongtao Li.

**Supervision:** Hongtao Li, Qiang Yao.

**Visualization:** Yu Wu.

**Writing – original draft:** Dengze Luo, Dong Li.

**Writing – review & editing:** Hongtao Li, Xingguo Yang, Qiang Yao.

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
