## [Decision Letter · Decision Letter 0]

22 Feb 2021

PONE-D-20-33037

Cloud Model-based Evaluation of Landslide Dam Development Feasibility

PLOS ONE

Dear Dr. Yao,

Thank you for submitting your manuscript to PLOS ONE. After careful consideration, we feel that it has merit but does not fully meet PLOS ONE’s publication criteria as it currently stands. Therefore, we invite you to submit a revised version of the manuscript that addresses the points raised during the review process.

Please note that the reviewer still has a few suggestions for revision, which need to be revised by the author.

We look forward to receiving your revised manuscript.

Kind regards,

Hanna Landenmark

Associate Editor

PLOS ONE

on behalf of 

Yiming Tang, Ph.D.

Academic Editor

PLOS ONE

Journal Requirements:

1. We suggest you thoroughly copyedit your manuscript for language usage, spelling, and grammar. If you do not know anyone who can help you do this, you may wish to consider employing a professional scientific editing service.  

2. Please ensure that all data sources are referenced within the manuscript and the Data availability statement.

3. We note that Figure 4 in your submission contains satellite images which may be copyrighted.

We require you to either (a) present written permission from the copyright holder to publish this figure specifically under the CC BY 4.0 license, or (b) remove the figure from your submission:

a. You may seek permission from the original copyright holder of Figure 4 to publish the content specifically under the CC BY 4.0 license. 

b. If you are unable to obtain permission from the original copyright holder to publish this figure under the CC BY 4.0 license or if the copyright holder’s requirements are incompatible with the CC BY 4.0 license, please either i) remove the figure or ii) supply a replacement figure that complies with the CC BY 4.0 license. Please check copyright information on all replacement figures and update the figure caption with source information. If applicable, please specify in the figure caption text when a figure is similar but not identical to the original image and is therefore for illustrative purposes only.

Additional Editor Comments (if provided):

Reviewers' comments:

Reviewer's Responses to Questions

**Comments to the Author**

1. Is the manuscript technically sound, and do the data support the conclusions?

Reviewer #1: Yes

Reviewer #2: Yes

Reviewer #3: Yes

2. Has the statistical analysis been performed appropriately and rigorously? 

Reviewer #1: Yes

Reviewer #2: Yes

Reviewer #3: Yes

3. Have the authors made all data underlying the findings in their manuscript fully available?

Reviewer #1: Yes

Reviewer #2: Yes

Reviewer #3: Yes

4. Is the manuscript presented in an intelligible fashion and written in standard English?

Reviewer #1: Yes

Reviewer #2: Yes

Reviewer #3: Yes

5. Review Comments to the Author

Reviewer #1: This paper mainly focuses on a special phenomenon formed under the condition of landslide geological disasters—landslide dam, and studies on the feasibility of landslide dam development. By establishing a top-down index system and comprehensively considering the randomness and ambiguity of development feasibility, the authors propose a cloud model improvement evaluation method based on cloud model uncertainty reasoning. And the improved method has been used in the analysis and evaluation of Hongshiyan and Tangjiashan landslide dams. The final results show that it has certain guiding significance for actual project management.

However, there are still some minor suggestions for this revised version. It could be published after minor revisions. The comments are as below.

1) In the abstract of the manuscript, the authors mentioned the principle of maximum membership to highlight the advantages of cloud model analysis. But throughout the full paper, the authors didn’t mention the basic theory, and analysis steps of maximum memberships, and there is no relevant literature cited. Therefore, it is suggested the authors can add something to this basic theory to make it clearer and easier for readers to understand.

2) In the introduction, the author mentioned “Xu et al. used a fuzzy mathematical method to evaluate the risk grade of the Hongshiyan landslide dam, and established six main indices for rating risk”. What are these six main indices for rating risk?

3) In the introduction, when the authors reviewed the literature, the research results of the cited scholars and the authors’ final summary do not have full correspondence. It is recommended that the authors may consider this issue. Discussions are needed.

4) In section 2.1, the authors mentioned the influencing factors of landslide dam development, and there are four aspects in total explained by the authors, including safety risk, resource feasibility, economic feasibility and ecological environmental impact. In order to highlight the importance of these four aspects, it is suggested the authors may consider using examples to explain these four points.

5) In section 3.1, the (a), (b), (c) of the Fig.1 are not in the same layout, which affects the look and understanding. The authors should correct this phenomenon. What’s more, the Tab.3 below the Fig.1 wasn’t cited.

6) In section 3.4, the authors mentioned “For the left and right boundaries, En was tripled to guarantee a membership of 0.5 at the comment division boundary, as shown in Tab. 8. See comment clouds in Fig. 2”. The Tab.8 should be changed to Tab.5, so that it corresponds to the table below.

7) There are many quoted figures missing in the paper, including Fig.2-Fig.8. That the authors should add them.

8)In section 5.2, the authors mentioned that the factors restricting the development of the Tangjiashan landslide dam are engineering safety risks and economic feasibility. Are there any countermeasures to reduce or avoid such risks?

9) It is suggested that the authors add contents to the conclusion, which are the measures for avoiding and reducing the unfavorable factors affecting the development of landslide dams.

10)The manuscript fully uses mathematical relationships to explain the weighting relationship of factors. Some qualitative and theoretical explanations can be added to make it more convincing.

Reviewer #2: The manuscript entitled “Cloud Model-based Evaluation of Landslide Dam Development Feasibility” presents the developmental feasibility of landslide dams with case studies from China. The research work has highlighted the importance of landslide dams as a potential resource for hydro-energy and tourism resources and determining their development feasibility is a great initiative towards water sustainability. The authors have conducted thorough data analysis for the development of a cloud model-improved evaluation method to evaluate the development feasibility of Hongshiyan and Tangjiashan landslide dams. With the successful application of improved model in the study the authors have been able to introduce the long-term management plan for the landslide dams. The authors have elaborated the evaluation methods with all the relevant technical details in a comprehensive manner. I would highly recommend this paper for publication so that other researchers in this field can benefit from such studies and play their part towards real time case studies in different parts of the world.

Reviewer #3: Dear Authors,

I admire your work. The findings and results are very interesting. The ideas are very well organized. The statistical approaches are adequate to find the solutions. The manuscript is very well written and meets the aims of it.

Thank you

6. PLOS authors have the option to publish the peer review history of their article (what does this mean?). If published, this will include your full peer review and any attached files.

Reviewer #1: No

Reviewer #2: No

Reviewer #3: No

---

## [Author Response · Author response to Decision Letter 0]

13 Mar 2021

Dear Editor and Reviewers,

We gratefully thank you for your time spend making the constructive remarks and useful suggestions, which has significantly raised the quality and has enabled us to improve the manuscript. Each suggested revision and comment brought forward by the reviewers was accurately incorporated and considered. Below the comments of the reviewers are response point by point and the revisions are indicated. 

Reviewer #1:

This paper mainly focuses on a special phenomenon formed under the condition of landslide geological disasters—landslide dam, and studies on the feasibility of landslide dam development. By establishing a top-down index system and comprehensively considering the randomness and ambiguity of development feasibility, the authors propose a cloud model improvement evaluation method based on cloud model uncertainty reasoning. And the improved method has been used in the analysis and evaluation of Hongshiyan and Tangjiashan landslide dams. The final results show that it has certain guiding significance for actual project management.

However, there are still some minor suggestions for this revised version. It could be published after minor revisions. The comments are as below.

1) In the abstract of the manuscript, the authors mentioned the principle of maximum membership to highlight the advantages of cloud model analysis. But throughout the full paper, the authors didn’t mention the basic theory, and analysis steps of maximum memberships, and there is no relevant literature cited. Therefore, it is suggested the authors can add something to this basic theory to make it clearer and easier for readers to understand.

Response: Thank you for your constructive and helpful suggestion. Following your suggestion, we have explained the theoretical of the maximum membership principle by adding the following text in the revised manuscript:

Section 5.1, paragraph 1, line 414, “The level corresponding to the element with the largest membership degree in the fuzzy evaluation vector is taken as the evaluation result, which is the maximum membership principle [29].”

2) In the introduction, the author mentioned “Xu et al. used a fuzzy mathematical method to evaluate the risk grade of the Hongshiyan landslide dam, and established six main indices for rating risk”. What are these six main indices for rating risk?

Response: We feel sorry for the inconvenience brought to the reviewer. The six main indices for rating risk now have been added into the manuscript as follows:

Section 1, paragraph 2, line 67, “Xu et al. used a fuzzy mathematical method to evaluate the risk grade of the Hongshiyan landslide dam, and established six main indices (i.e., social development; dam material, volume parameters, water level growth rate, mountain stability and river channel river) for rating risk [1].”

3) In the introduction, when the authors reviewed the literature, the research results of the cited scholars and the authors’ final summary do not have full correspondence. It is recommended that the authors may consider this issue. Discussions are needed.

Response: Thank you for your rigorous consideration. In the introduction, we cited scholars’ views “Studies have pointed out that the development of a landslide dam must ensure its safety and health, and consider environmental compatibility, social benefits, and economic benefits”，while in the conclusion, we evaluate the landslide dam development feasibility from four aspects, i.e., safety risk, resource feasibility, economic feasibility, and eco-environmental impact. Following the reviewer’s comment, the inconsistency of the indicators is discussed as follows:

Section 5.2, paragraph 2, line 465, “On the contrary, if the landslide dam is retained for further development, a built-up landslide dam could not only create tourism, irrigation, electricity generation, and other economic benefits [8], but also increase the social benefits, promoting the local socio-economic development. Therefore, despite the relatively low economic feasibility of the Tangjiashan landslide dam, its potential national social benefits are still considerable. Considering the Tangjiashan landslide dam hasn’t been developed, it is hard to measure its social benefits currently, thus in this paper, we only choose economic benefits as evaluation indicator, a more comprehensive evaluation of the landslide dam development feasibility can be conducted combining with social benefits in the future”

If there is any other inconsistency between the introduction and conclusion that we don’t notice, we sincerely hope you could point it out to us.

4) In section 2.1, the authors mentioned the influencing factors of landslide dam development, and there are four aspects in total explained by the authors, including safety risk, resource feasibility, economic feasibility and ecological environmental impact. In order to highlight the importance of these four aspects, it is suggested the authors may consider using examples to explain these four points.

Response: Thank you for your valuable suggestion. An example of landslide dam development feasibility evaluation is added in the manuscript as follows:

Section 2.1, paragraph 1, line 106, “The indices constituting a feasibility evaluation system must be able to reflect their overall characteristics and influencing factors, while being independent of each other and easy to obtain. Decisions about landslide dam development and utilization are influenced by multiple factors. Taking Hongshiyan landslide dam as an example, emergency measures are taken to reduce the breaching probability after its formation, and the potential hydropower energy, economic benefits and other conditions are taken into account to evaluate the potential for development based on the safety of the landslide dam [8]. So, referring ……”

5) In section 3.1, the (a), (b), (c) of the Fig.1 are not in the same layout, which affects the look and understanding. The authors should correct this phenomenon. What’s more, the Tab.3 below the Fig.1 wasn’t cited.

Response: We appreciate for your valuable comment. The layout of these three figures is adjusted as follows:

(a) Forward cloud generator

(b) Backward cloud generator

(c) Conditional cloud generator

Fig. 1 Three kinds of Cloud generator

The Tab. 3 was cited at the end section 3.1, which is shown as follows:

Section 3.1, paragraph 2, line 223, “Tab. 3 shows the basic operations of cloud.”

Section 3.5, paragraph 1, line 316, “(4) Synthesize the operation formula and determine the membership S through evaluation based on cloud operation rules (Tab 3);”

6) In section 3.4, the authors mentioned “For the left and right boundaries, En was tripled to guarantee a membership of 0.5 at the comment division boundary, as shown in Tab. 8. See comment clouds in Fig. 2”. The Tab.8 should be changed to Tab.5, so that it corresponds to the table below.

Response: Thank for your careful review. The mistake is modified in this manuscript as follows. We feel sorry about our carelessness.

Section 3.4, paragraph 1, line 298, “For the left and right boundaries, En was tripled to guarantee a membership of 0.5 at the comment division boundary, as shown in Tab. 5. See comment clouds in Fig. 2.”

7) There are many quoted figures missing in the paper, including Fig.2-Fig.8. That the authors should add them.

Response: Thank you for your suggestion. As suggested by reviewer, we have added the suggested figures to the manuscript.

8) In section 5.2, the authors mentioned that the factors restricting the development of the Tangjiashan landslide dam are engineering safety risks and economic feasibility. Are there any countermeasures to reduce or avoid such risks?

Response: We are grateful for your comment. The countermeasures to reduce such risks are added in the manuscript in the section 5.2, showing as follows:

Section 5.2, paragraph 2, line 479, “Thus, if the strategy is to retain the Tangjiashan landslide dam, engineering measures should be taken to ensure its safety. The countermeasures should focus on reducing the risk of landsliding, piping and overtopping of the dam. Vibrating compaction can be used to improve dam stability, combining with the slope revetment in the upstream and rock pile pressure on the slope toe of the downstream on the basis of knowledge of the physical properties of dam materials. Grouting engineering measures can deal with potential piping issues. For example, an engineering sealant measure was applied to the Xiaonanhai landslide dam with excellent sealant effectiveness according to leakage observation in later periods [8]. Besides, widening and lining the existed spillway combining with setting crown wall on the dam top are useful to prevent overtopping. Factors affecting the development of the landslide dam can also be reduced by increasing the downstream flood protection capacity through the construction of facilities such as flood barriers.”

9) It is suggested that the authors add contents to the conclusion, which are the measures for avoiding and reducing the unfavorable factors affecting the development of landslide dams.

Response: Thank for your suggestion. Following your advice, we have added the measures for avoiding and reducing the unfavorable factors affecting the development of landslide dams in the conclusion as follows:

Section 6, paragraph 4, line 536, “The Hongshiyan landslide dam evaluation showed relatively high development feasibility, which now has been developed and putted into use. While the Tangjiashan landslide dam had low development feasibility, the factors restricting its development are engineering safety risks and economic feasibility, engineering measures like grouting, vibrating compaction and lining the existed spillway can be applied to reducing the safety risks, so as to turn disasters into benefits to mankind.”

10) The manuscript fully uses mathematical relationships to explain the weighting relationship of factors. Some qualitative and theoretical explanations can be added to make it more convincing.

Response: We are extremely grateful to reviewer for pointing out this problem. We have added some qualitative explanations of the weighting result in the section 4.1.3, which is shown as follows:

Section 4.1.3, paragraph 1, line 352, “From the aspect of the indicator layer, the safety risk has the greatest weight, which reflects the opinion of safety is the basis of the landslide dam development [20]. While the weight of eco-environmental impact is bigger than the economic feasibility and resource feasibility, indicating that the environmental protection plays a more critical role in the development of landslide dam. The close weighting of economic feasibility and resource feasibility indicates that they are of similar importance.”

We sincerely hope that this revised manuscript has addressed all your comments and suggestions. We appreciated for the reviewers’ warm work earnestly, and hope that the correction will meet with approval. Once again, thank you very much for your comments and suggestions.

---

## [Decision Letter · Decision Letter 1]

22 Apr 2021

Cloud Model-based Evaluation of Landslide Dam Development Feasibility

PONE-D-20-33037R1

Dear Dr. Yao,

We’re pleased to inform you that your manuscript has been judged scientifically suitable for publication and will be formally accepted for publication once it meets all outstanding technical requirements.

Kind regards,

Yiming Tang, Ph.D.

Academic Editor

PLOS ONE

Additional Editor Comments (optional):

Reviewers' comments:

Reviewer's Responses to Questions

**Comments to the Author**

1. If the authors have adequately addressed your comments raised in a previous round of review and you feel that this manuscript is now acceptable for publication, you may indicate that here to bypass the “Comments to the Author” section, enter your conflict of interest statement in the “Confidential to Editor” section, and submit your "Accept" recommendation.

Reviewer #1: (No Response)

Reviewer #2: All comments have been addressed

2. Is the manuscript technically sound, and do the data support the conclusions?

Reviewer #1: Yes

Reviewer #2: Yes

3. Has the statistical analysis been performed appropriately and rigorously? 

Reviewer #1: Yes

Reviewer #2: Yes

4. Have the authors made all data underlying the findings in their manuscript fully available?

Reviewer #1: Yes

Reviewer #2: Yes

5. Is the manuscript presented in an intelligible fashion and written in standard English?

Reviewer #1: Yes

Reviewer #2: Yes

6. Review Comments to the Author

Reviewer #1: (No Response)

Reviewer #2: (No Response)

7. PLOS authors have the option to publish the peer review history of their article (what does this mean?). If published, this will include your full peer review and any attached files.

Reviewer #1: No

Reviewer #2: No

---

## [Editor Report · Acceptance letter]

28 Apr 2021

PONE-D-20-33037R1 

Cloud Model-based Evaluation of Landslide Dam Development Feasibility 

Dear Dr. Yao:

I'm pleased to inform you that your manuscript has been deemed suitable for publication in PLOS ONE. Congratulations! Your manuscript is now with our production department. 

Kind regards, 

on behalf of

Professor Yiming Tang 

Academic Editor

PLOS ONE